# Hepatocellular Carcinoma with Bile Duct Tumor Thrombus: A Case Report and Literature Review of 890 Patients Affected by Uncommon Primary Liver Tumor Presentation

**DOI:** 10.3390/jcm12020423

**Published:** 2023-01-04

**Authors:** Maria Conticchio, Nicola Maggialetti, Marco Rescigno, Maria Chiara Brunese, Roberto Vaschetti, Riccardo Inchingolo, Roberto Calbi, Valentina Ferraro, Michele Tedeschi, Maria Rita Fantozzi, Pasquale Avella, Angela Calabrese, Riccardo Memeo, Arnaldo Scardapane

**Affiliations:** 1Unit of Hepatobiliary Surgery, Miulli Hospital, 70124 Acquaviva Delle Fonti, Italy; 2Interdisciplinary Department of Medicine, Section of Radiology and Radiation Oncology, University of Bari “Aldo Moro”, 70124 Bari, Italy; 3Radiology Unit, Miulli Hospital, 70124 Acquaviva Delle Fonti, Italy; 4Clinical Pharmacology Unit, A. Cardarelli Hospital, 86100 Campobasso, Italy; 5Department of Clinical Medicine and Surgery, “Federico II” University of Naples, 80131 Naples, Italy; 6Radiology Unit–IRCCS “Giovanni Paolo II”, 70124 Bari, Italy

**Keywords:** HCC, hepatobiliary surgery, bile duct tumor thrombus, BDTT

## Abstract

Bile duct tumor thrombus (BDTT) is an uncommon finding in hepatocellular carcinoma (HCC), potentially mimicking cholangiocarcinoma (CCA). Recent studies have suggested that HCC with BDTT could represent a prognostic factor. We report the case of a 47-year-old male patient admitted to the University Hospital of Bari with abdominal pain. Blood tests revealed the presence of an untreated hepatitis B virus infection (HBV), with normal liver function and without jaundice. Abdominal ultrasonography revealed a cirrhotic liver with a segmental dilatation of the third bile duct segment, confirmed by a CT scan and liver MRI, which also identified a heterologous mass. No other focal hepatic lesions were identified. A percutaneous ultrasound-guided needle biopsy was then performed, detecting a moderately differentiated HCC. Finally, the patient underwent a third hepatic segmentectomy, and the histopathological analysis confirmed the endobiliary localization of HCC. Subsequently, the patient experienced a nodular recurrence in the fourth hepatic segment, which was treated with ultrasound-guided percutaneous radiofrequency ablation (RFA). This case shows that HCC with BDTT can mimic different types of tumors. It also indicates the value of an early multidisciplinary patient assessment to obtain an accurate diagnosis of HCC with BDTT, which may have prognostic value that has not been recognized until now.

## 1. Introduction

Colorectal liver metastases are the most frequent secondary liver tumor; on the other hand, hepatocellular carcinoma (HCC) is the most common primary liver cancer, and surgical resection is the mainstay treatment [1,2,3,4,5,6,7].

HCC with bile duct tumor thrombus (BDTT) is relatively uncommon, with an incidence of 0.5–12.9% [8].

HCC with BDTT was first described by Mallory et al. in 1947, and in 1975, by Lin et al. named BDTT “Icteric-Type Hepatocarcinoma” based on the often-associated symptom of jaundice, although it may not be obvious at first diagnosis [9].

Patients with a history of HBV or HCV infection have a higher risk of developing BDTT in the background of viral damage to the liver parenchyma. The way in which BDTT develops is controversial, but can be summarized into two main pathways: cancer cells can compose the thrombus or they can cause a cancerous thrombosis due to blood clots consequent an invasive hemorrhage of the bile duct wall [9].

The mechanisms of these two pathways are not yet known, but four hypotheses have been formulated. Firstly, cancer cells found inside the bile duct can be directly related to the primary tumor, which expands itself until invading the bile duct and creating a thrombosis [9]. Secondly, the primary tumor can invade microvessels and lymphatic vessels, invading the micro-circle of the bile duct entering the biliary system [9]. Thirdly, it can be hypothesized that the creation of an arteriovenous shunt to the bile duct system might be a way of tumor cells diffusion [9]. The last hypothesis pertains to the cells’ diffusion through the nerves that cover the wall of the bile duct, but there are not much evidences of this behavior [9]. In clinical practice, nowadays, the most reliable hypothesis is the first one, because even micro-BDTT is considered a feature of the invasiveness of HCC and it is related to a poorer prognosis. A new relationship between micro-BDTT and the inflammatory pathway has been described [10].

Despite the evolving technologies in imaging diagnosis and the support of Artificial Intelligence in Hepatobiliary and Pancreatic Surgery (HPB) [5,11,12,13,14], the diagnosis of HCC with BDTT is very challenging. Its treatment is still debated due to a complex pre- and postoperative patients’ management [15,16].

HCCs with BDTT are always associated with both parenchymal and intraductal lesions [17], but the primary tumor can generate a thrombus when the parenchymal lesion is still small and undetectable by preoperative imaging [9,17,18].

So the incidence of BDTT without macroscopic HCC as a specific subtype is not really evident in the literature, but there are no doubts that it might further jeopardize the diagnosis of HCC with BDTT [8].

Furthermore, another challenge concerning HCC with BDTT is represented by the clinically and radiologically mimetism with cholangiocarcinoma (CCA) [8,17].

Laboratory and clinical-anamnestic data can help the differential diagnosis: predisposing factors for cirrhosis such as hepatitis B (HBV) or C (HCV) virus and elevated serum α-fetoprotein (AFP) levels may suggest HCC with BDTT diagnosis [19].

CCAs may include cholestasis (such as primary sclerosing cholangitis, hepatolithiasis, or bile duct cysts) and chronic inflammation pathway (such as biliary parasitosis, viral hepatitis, or Non-Alcoholic SteatoHepatitis (NASH)) [20].

Carbohydrate antigen 19-9 (CA 19-9) and carcinoembryonic antigen (CEA), as preoperative diagnostic biomarkers of CCAs, showed low sensitivity [21].

Integration with imaging features and, if necessary, with biopsy histological reports is still required to make the final diagnosis.

So considering the challenging clinical picture and the lack of evidences in literature, we aim to clearly describe the management of a HCC with BDTT compared with most relevant experiences already reported. 

## 2. Case Report

A 47-year-old male patient was admitted to the University Hospital Policlinico of Bari (Italy) with abdominal pain in the right hypochondrium.

During the initial examination, the patient incidentally tested HBV-positive, in the absence of jaundice. Subsequent blood tests revealed an untreated HBV infection and normal liver and pancreatic function tests.

The laboratory findings were HBsAg 5000 IU/mL (normal value (NV) < 0.05 IU/mL); AFP 2157 ng/mL (NV 0–5 ng/mL); total bilirubin 0.9 mg/dL (NV 0.3–1 mg/dL), direct bilirubin 0.2 mg /dL (NV 0–0.4 mg/dL), indirect bilirubin 0.7 mg/dL (NV 0.1–1 mg/dL), γ-GT 44 U/l (NV < 50 U/L), and CA19–9 222 U/mL (NV 0–37 U/mL); the CEA level was undetectable.

An emergency abdominal ultrasound showed inhomogeneity hepatic echo structure. The bile duct of the third segment showed segmental dilation, furthermore the gallbladder appeared distended, without gallstones and regular wall.

Subsequently, an upper-abdomen Computed Tomography (CT) scan and liver Magnetic Resonance Imaging (MRI) were performed (Figure 1, Figure 2, Figure 3, Figure 4, Figure 5, Figure 6 and Figure 7).

A mass of 1.8 cm was detected within the biliary branch for the third hepatic segment, characterized by nodular impregnation in the arterial phase and irregular and partial washout in the portal venous phase. These findings were compatible with a heterologous lesion, although it was not possible to perform a diagnosis among HCC, intrahepatic CCA or other pathological lesions. 

Consequently, in order to obtain a histological diagnosis, a percutaneous ultrasound-guided needle biopsy was performed.

The histological report revealed the presence of a moderately differentiated HCC (Edmondson grade II); immunohistochemistry results showed positive CK7 and CD34 staining and negative HEP par-1 staining consistent with the hypothesis.

Therefore, after a multidisciplinary team (MDT) discussion, third segment segmentectomy was performed through a minimally invasive approach, with intraoperative examination confirming endoluminal HCC with BDTT.

The histopathological analysis of the surgical specimen confirmed endobiliary metastasis from HCC. The postoperative histological report showed hepatic cirrhosis, end-stage HBV-related, and dysplastic nodules. The invasion of the major hepatic ducts was caused by carcinomatous proliferation with the morphological features of HCC.

The postoperative course was uneventful, and the patient was discharged on postoperative day 7.

No recurrence was evidenced until 6 months postoperatively, while a CT scan detected a nodule with HCC typical radiological characteristics (~1 cm in greatest diameter) in the fourth segment, with AFP serum level of 280 ng/mL. MDT team decide to perform ultrasound-guided percutaneous radiofrequency ablation (RFA) of the nodule. A CT scan carried out 1 month after ablation confirmed the nodule’s complete necrosis.

## 3. Discussion

Our experience showed how challenging is the diagnosis of HCC with BDTT and its impact on further management of the patients. Another key point is absence of a systematic classification that includes BDTT as a prognostic factor. These two points do not allow clinicians to appropriately relate BDTT to a stage of HCC; however, surgical treatment appears to be the first treatment option [22,23,24,25,26,27,28,29]. Considering the undeniable benefits of minimally invasive surgery, we have to underline that it offers a safer surgical approach for patients with a Performance Status (PS) of 1 or 2, allowing a shorter length of stay and faster recovery which as show a great impact also in patients undergoing surgical downstaging [22,23,24,25,26,27,28,29].

Furthermore, the COVID-19 pandemic era changed the surgical scenario of specialized surgical fields such as HPB surgery. A huge number of Hub and Spoke learning programs allowed the peripheral center to achieve high specialization in HPB [2]. During the pandemic, these programs always granted the standard of care for patients with consequent savings of time and money, avoiding the costs of health mobility [30].

Besides the diffusion of the newest surgical skills, it was necessary to support them with the earliest accurate diagnosis of cancer. This is probably the most important effort to improve patients’ survival. In oncological imaging, CT remains the main diagnostic tool for detection, follow-up and tumor staging [31].

Therefore, the achievement of the most precise treatment for each patient requires the diffusion of standardized diagnostic protocols [32,33].

To better understand the results of our case, we performed literature research through the main search engines (PubMed and Medline).

Regarding our literature review, we have extrapolated 20 articles with a total population of 890 patients [8,9,17,34,35,36,37,38,39,40,41,42,43,44,45,46,47,48,49,50] (Table 1 and Table 2).

Concerning baseline characteristics, our patient is 47 years old, which is in line with the literature mean age (55.35 ± 9.49). As shown in Table 1, males are more frequently affected by this pathology than females (82.80% vs. 17.20%) [8,9,17,34,35,36,37,38,39,40,41,42,43,44,45,46,47,48,49,50].

Jaundice is the most common presenting symptom, as reported by 13 of 20 articles. A total of 53.28% (317/595) of patients had jaundice as a presenting symptom (Table 1).

The 57.97% (200/345) patients showed a mean AFP value >400 ng/mL (Table 2).

A meta-analysis conducted by Navadgi et al. in 2016 compared clinic-pathological characteristics and survival outcomes between HCC patients who underwent hepatic resection, with and without BDTT, including 6,051 patients from 11 studies, mostly conducted in Asia [51].

Patients with HCC with BDTT had worse histological features compared to those without BDTT in terms of higher rates of macrovascular and lymphovascular invasion and poorer differentiation. However, in the BDTT group after hepatectomy, this meta-analysis revealed an inferior long-term survival rate, with no decrease in the 3-year survival rate [51].

Another retrospective analysis, conducted by Wong et al., compared outcomes between all 37 HCC patients with BDTT and 222 control patients who underwent hepatic resection between 1989 and 2012. Notably, it also revealed similar 5-year overall survival (OS) and disease-free survival (DFS) when matched for tumor stage and adverse prognostic factors, which seems to suggest that BDTT was not relevant for patients’ prognosis [34].

So BDTT is not included as a prognostic factor in the most common HCC staging systems, such as Barcelona Clinic Liver Cancer (BCLC) and American Joint Commission on Cancer (AJCC) [52,53].

Anyway, recent evidences are still debating on the topic.

A retrospective study by Lu et al. analyzed 622 HCC Chinese patients who underwent hepatic resections, considering that the BCLC staging system is mainly based on data from Western HCC populations. The most commonly underlying liver disease was HBV (77%). This study revealed that patients with HCC with BDTT had a worse OS at 1, 3, and 5 years compared to those without BDTT (77%, 42%, and 23% vs. 80%, 60%, and 48%, respectively), limited, however, to the early stages of the disease (BCLC 0 and BCLC A). After recategorizing HCC with BDTT 0-A as BLCL B, the modified BLCL staging system showed a better prediction of OS and mortality [54].

In our review, 485 (485/838 57.87%) patients had a positive HBV test [8,9,17,34,35,36,37,38,39,40,41,42,43,45,46,47,48,50] (Table 1).

1-, 3-, and 5-year OS are reported in Table 2 [34,35,36,38,40,42,43,44,45,46,47,48,49].

Huang et al. analyzed outcomes of 1021 patients with HCC underwent R0 resection at 9 hepatobiliary referral centers. A total of 177 (17.34%) presented BDTT and it seems to be an independent risk factor. Furthermore, HCC with BDTT without macrovascular invasion was classified as BCLC B and AJCC IIIA, whereas HCC with BDTT with macrovascular invasion was classified as BCLC C and AJCC IIIB [55].

In addition, an higher incidence of post-liver-transplant recurrence in HCC patients with BDTT has been reported in the literature, although not in large-volume studies [35].

These results appear consistent with our clinical experience.

A single lesion ≤ 2 cm in diameter, such as the one we have described, with preserved liver function, is currently staged as BCLC 0, whereas nodular recurrence, which occurred in our case report, appears to be more consistent with a worse prognostic pattern.

HCC with BDTT is both clinically and radiologically difficult to distinguish from other primary biliary cancers, especially CCA.

CCAs are divided by anatomical localization into three types: perihilar, intrahepatic, and peripheral. This classification has also a prognostic and therapeutic value [56].

Perihilar CCA or Klatskin tumor is the most common one. Its growth is more often of the “periductal-infiltrating” type. It tends to be diagnosed earlier, with a smaller size compared to the intrahepatic one, due to the earlier presentation of symptoms.

Intrahepatic CCA more often has “mass-forming” growth, well limited from the surrounding hepatic parenchyma [57].

It is the most common type in the absence of other tumors or cirrhosis, although it can coexist with such diseases.

Peripheral CCA has histological features similar to the perihilar type [56].

A misdiagnosis between HCC with BDTT and CCA is described with an incidence of 4–55% [17,36].

However, it is important to recognize both of them in order to define patient management.

CCA patients are usually not candidates for liver transplantation because, despite radical surgery, they recur in 60% of cases, mainly in the first 2 years [56,58].

Surgery for CCA, if resectable, varies based on the location. For example, perihilar CCAs are usually treated with the resection of the biliary convergence with the main biliary duct, with major hepatic resection and caudectomy, while peripheral CCAs are treated with pancreatic-duodenectomy [56].

Conversely, partial hepatic resection or hemihepatectomy with bile duct preservation is the main surgical option for HCC with BDTT [17,59]. Recent Asian studies have proposed a more aggressive surgical approach, including major liver resection combined with bile duct resection [36].

HCC with hilar bile duct tumor thrombus (HBDTT) is a common subtype of HCC with BDTT, and it shares some imaging features with perihilar CCA: hilar mass, obstructed hilar bile duct, and upstream bile duct dilatation [17].

However, some other features can help in the differential diagnosis.

HCC typically has an increased arterial blood supply, so it usually shows hyperattenuation in the arterial phase and hypoattenuation in the portal venous phase, compared to the hepatic parenchyma.

Most HBDTTs should show the same enhancement pattern. However, some HCCs can show iso- or hypoattenuation in the arterial phase, with the enhancement in the arterial phase inversely correlated with the degree of blood clots and necrosis. So, hypoattenuation in the portal venous phase seems to be the most important imaging feature to distinguish HCC with BDTT from perihilar CCA [17].

Furthermore, HBDTT rarely infiltrates into the bile duct wall, which, consequently, is often regular without relevant enhancement.

Conversely, perihilar CCA more frequently reproduces the “periductal-infiltrating” type, so it usually shows a narrowed hilar bile duct with irregular or even obliterated wall thickening, with typically progressively delayed enhancement.

Washout in the portal venous phase is also the main feature to distinguish HCC with BDTT from intrahepatic CCA, together with the presence of tortuous tumoral vessels [17].

## 4. Conclusions

In conclusion, the diagnosis of HCC with BDTT can be reasonably considered in the presence of lesions of both hepatic parenchyma and bile ducts with a cirrhotic underlined liver disease, especially if they show typical washout in the portal venous phase.

However, several factors can jeopardize the diagnosis.

Furthermore, as already pointed out, the primary parenchymal tumor can be undetectable by preoperative imaging once the thrombus appears. These findings support the value of an early multidisciplinary patient assessment to obtain an accurate diagnosis of HCC with BDTT, which may have prognostic value that has not been recognized until now.

## Figures and Tables

**Figure 1 jcm-12-00423-f001:**
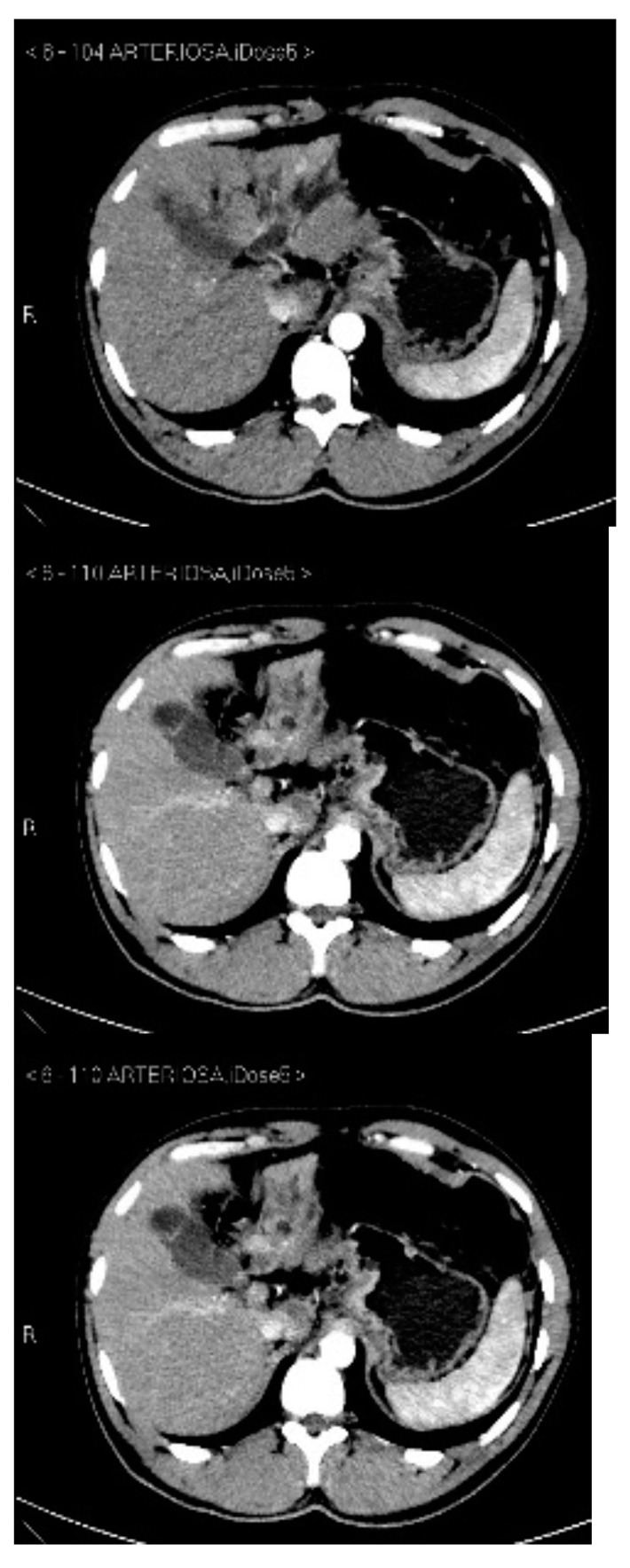
CT arterial phase.

**Figure 2 jcm-12-00423-f002:**
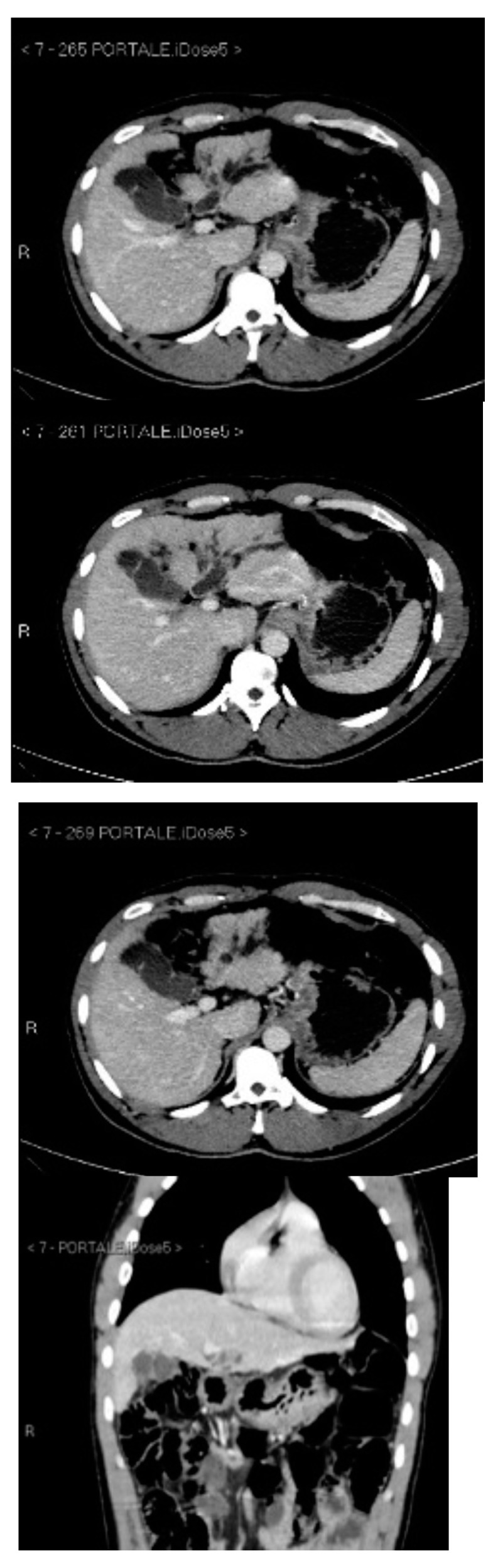
CT portal phase.

**Figure 3 jcm-12-00423-f003:**
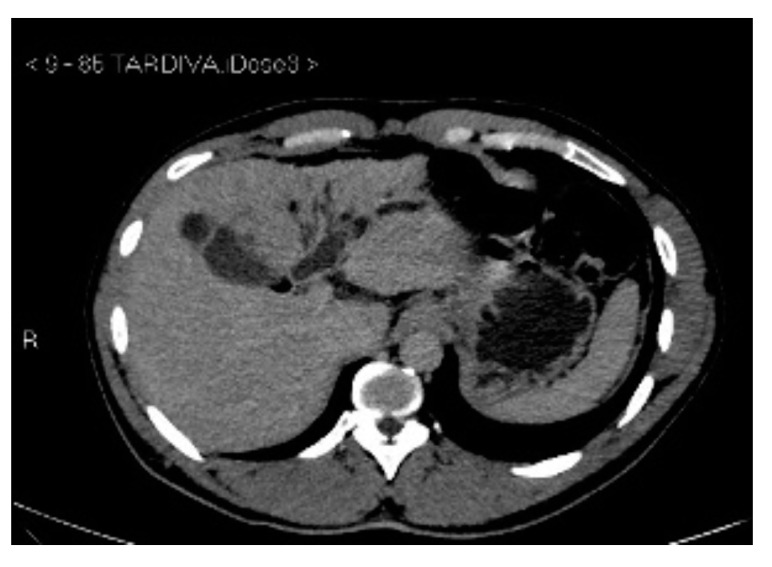
CT delayed phase.

**Figure 4 jcm-12-00423-f004:**
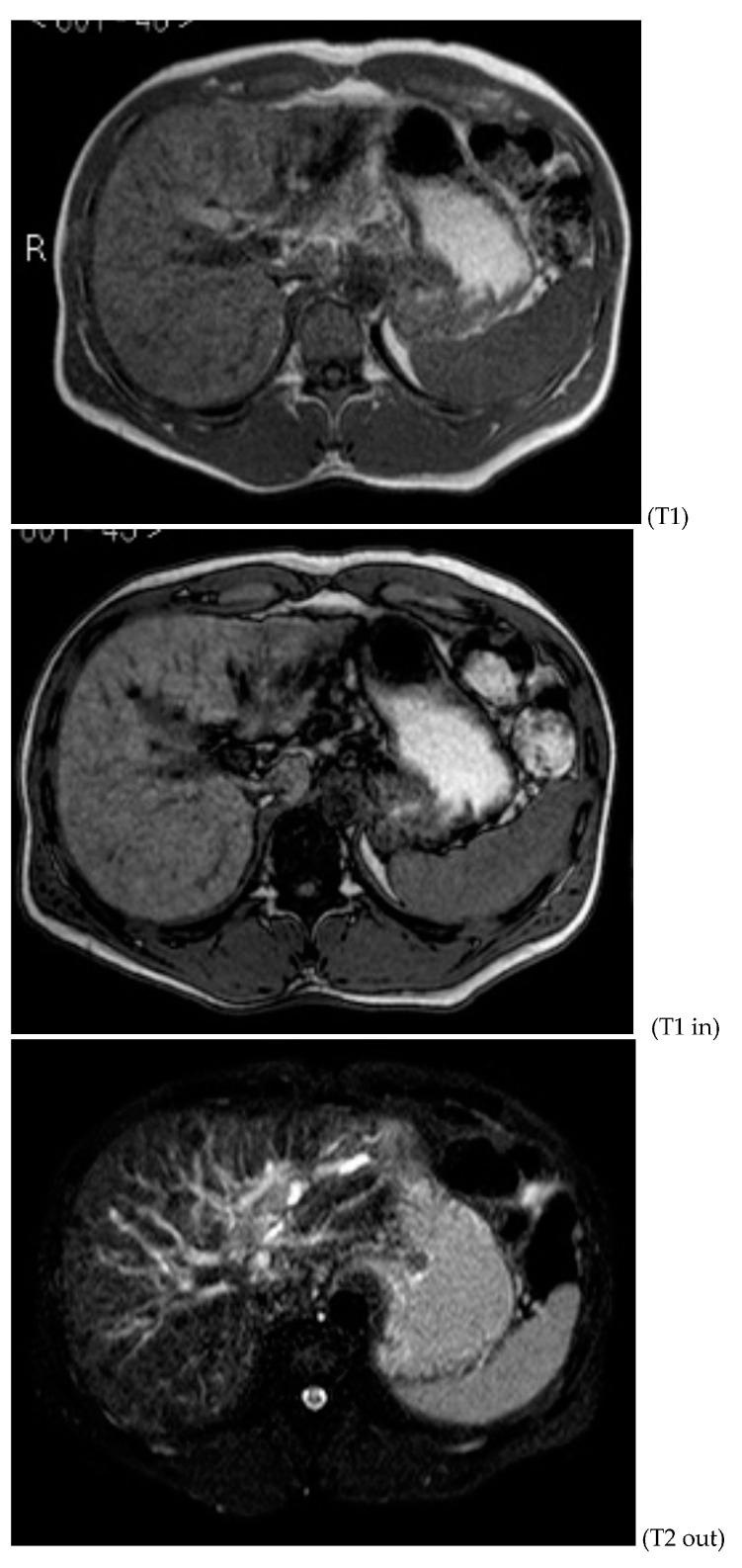
MRI T1, T1 in, T1 out, and DWI.

**Figure 5 jcm-12-00423-f005:**
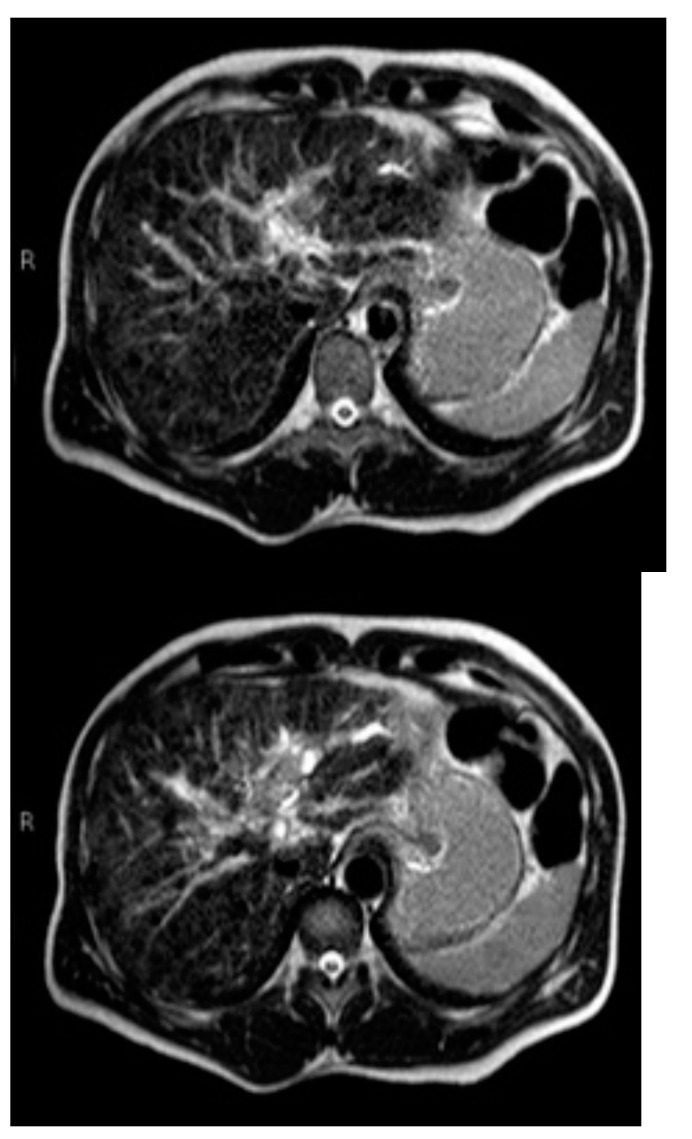
MRI T2.

**Figure 6 jcm-12-00423-f006:**
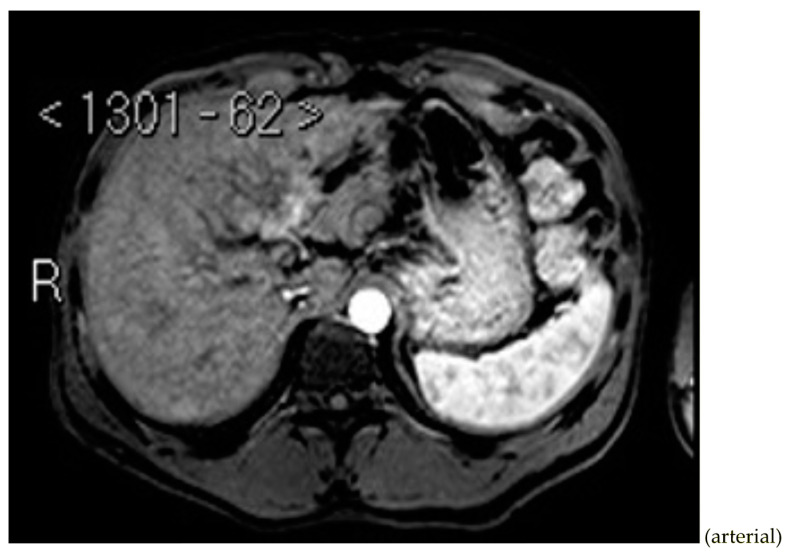
MRI arterial, portal, and HPB phases.

**Figure 7 jcm-12-00423-f007:**
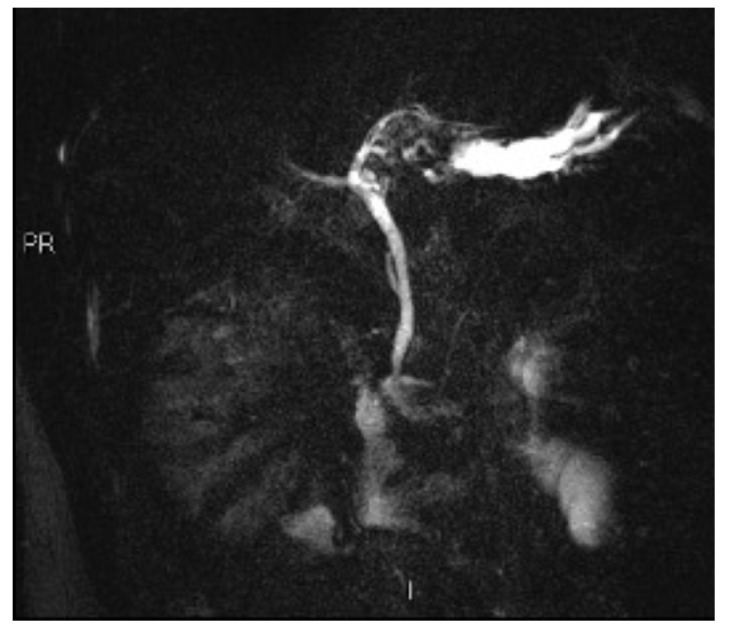
Cholangio-MRI.jpg.

**Table 1 jcm-12-00423-t001:** Literature review: baseline characteristics.

Author	Year	Study Type	N. of Cases	Age, Years	Sex, M/F	Symptoms	Jaundice	HBV Positive	Diagnosis
Satoh et al. [31]	2000	Retrospective cohort study	17	58.18 ± 8.94	15 (88.24)/2 (11.76)	Jaundice	9 (52.94)	5 (29.4)	Ultrasonography; CT
Shiomi et al. [38]	2001	Retrospective cohort study	17	58.8 ± 2	15 (88.24)/2 (11.76)	Jaundice, abdominal pain, poor appetite, general fatigue, or fever	10 (58.82)	7 of 14 (50)	Ultrasonography; CT
Peng et al. [39]	2004	Retrospective cohort study	8	51.75 ± 8.15	7 (87.5)/1 (12.5)	Jaundice	8 (100)	6 (75)	Ultrasonography; CT; MRI
Esaki et al. [40]	2005	Retrospective cohort study	19	59.79 ± 11.26	19 (100)/0 (0)	Jaundice, fever, or abdominal pain	NA	8 (42.10)	Ultrasonography; CT; MRI; angiography
Shao et al. [41]	2011	Retrospective cohort study	27	47.1 ± 10.5	23 (85.18)/4 (14.81)	NA	NA	8 (42.10)	Chest XR; abdominal ultrasonography; CT; CPRE
Yu et al. [42]	2011	Retrospective cohort study	20	50.6 ± 2.4	17 (85)/3 (15)	Obstructive jaundice and upper abdominal pain	14 (70)	16 (80)	NA
Noda et al. [43]	2011	Retrospective cohort study	22	45% were ≤60 y; 55% were >60 y	21 (95)/1 (5)	NA	8 (36.36)	15 (68.18)	Ultrasonography; CT; angiography; ERCP or MR cholangiopancreatography
Moon et al. [44]	2012	Retrospective cohort study	73	54.2 ± 11.1	52 (71.23)/21 (28.77)	Jaundice	34 (46.60)	59 (80.82)	CT; MRI
Oba et al. [45]	2014	Retrospective cohort study	13	60.85 ± 8.64	12 (92.31)/1 (7.69)	NA	NA	4 (30.77)	Ultrasonography; CT; MRI
Wong et al. [34]	2014	Retrospective cohort study	37	56.75 ± 14.75	29 (78.38)/8 (21.62)	NA	NA	30 (81.1)	CT; MRI
Rammohan et al. [46]	2014	Retrospective cohort study	39	52.1 ± 10.9	28 (71.80)/11 (28.20)	Jaundice	18 (46.10)	7 (17.9)	Abdominal ultrasonography; abdominal CT
Ha et al. [35]	2014	Retrospective cohort study	14	54.6 ± 5.6	10 (71.43)/4 (28.57)	Jaundice	9 (64.29)	11 (78.57)	NA
Kasai et al. [47]	2015	Retrospective cohort study	44	64 ± 9.1	35 (79.5) /9 (20.5)	Jaundice	27 (61.36)	8 (18.2)	NA
Chotirosniramit et al. [48]	2017	Retrospective cohort study	19	51.1 ± 11.5	15 (78.95)/4 (21.05)	Jaundice or cholangitis	14 (73.68)	16 (84.2)	Abdominal CT
Kim et al. [36]	2018	Retrospective cohort study	257	61 ± 11.6	210 (81.71)/47 (18.29)	Jaundice	120 (46.70)	115 (44.75)	NA
Lin et al. [49]	2019	Retrospective cohort study	49	55.51 ± 13.09	43 (87.75)/6 (12.25)	NA	NA	NA	Abdominal ultrasonography; abdominal CT; hepatic angiography; MR cholangiopancreatography
Zhou et al. [9]	2020	Retrospective cohort study	7	66 ± 6.24	6 (85.71)/1 (14.28)	Jaundice	7 (100)	4 (57.14)	CT; MRI;
Zhou et al. [17]	2020	Retrospective cohort study	58	49.84 ± 10.23	51 (87.93)/7 (2.07)	Jaundice and upper abdominal pain	39 (67.24)	42 (72.41)	CT
Sun et al. [50]	2020	Retrospective multicenter study	120	50.55 ± 11.35	106 (88.33)/14 (11.67)	NA	NA	95 (79.17)	NA
Wu et al. [8]	2021	Multicenter study	30	48.5 ± 13.04	23 (76.67)/7 (23.33)	NA	NA	29 (96.67)	CT; MRI
Conticchio et al.	2022	Case report	1	47	1 (100)/0 (0)	Abdominal pain	0 (0)	1 (100)	Ultrasonography; CT; MRI

Abbreviations: M, male; F, female; HBV, hepatitis B virus; CT, Computed Tomography; MRI, Magnetic Resonance Imaging; ERCP, Endoscopic Retrograde Cholangiopancreatography; NA, Not Available.

**Table 2 jcm-12-00423-t002:** Literature review: pre-, intra-, and postoperative characteristics.

Author	AFP (>400 ng/mL), *n*.	AFP, Mean ± SD	Total Bilirubin, Mean ± SD	Tumor Size, cm	Surgical Procedure	Mortality, *n*	Overall Survival, %		
							1-Year	3-Year	5-Year
Satoh et al. [31]	NA	NA	NA	NA	NA	12 (70.59)	NA	NA	NA
Shiomi et al. [38]	NA	73.87 ± 72.89	6.9 ± 1.8 mg/dL	6.1 ± 1.2	1 right hepatic trisegmentectomy with caudate lobectomy; 5 right hepatic lobectomies with caudate lobectomy; 6 left hepatic lobectomies with caudate lobectomy; 1 right anterior segmentectomy; 1 right anterior segmentectomy with caudate lobectomy; 1 segmentectomy S5; 1 S4; 1 S1	11 (64.70)	NA	47	28
Peng et al. [39]	7 (87.5)	NA	Ra 68.4–436.4 umol/L	Ra 2–9	3 hepatectomies with removal of the tumor thrombus; 1 hepatectomy combined with extrahepatic bile duct resection; 3 thrombectomies through a choledochotomy; 1 orthotopic liver transplantation	7 (87.5)	62.5	37.5	NA
Esaki et al. [40]	NA	NA	5 ± 7.3 mg/dL	NA	7 left hepatectomies; 1 lateral segmentectomy; 6 right hepatectomies; 1 central bisegmentectomy; 3 medial segmentectomies; 1 anterior segmentectomy	NA	79	45	33
Shao et al. [41]	16 (59.3)	NA	116.4 ± 135.4 umol/L	NA	1 right anterior resection; 2 right posterior resections; 4 right hepatectomies; 8 left hepatectomies; 1 left hepatectomy with caudate lobectomy; 3 left lateral resections; 2 left medial resections; 6 partial resections	1 (3.70)	NA	NA	NA
Yu et al. [42]	9 (45)	2651.85 ± 6135.32	123.25 ± 142.06 mol/L	3.65 ± 2.4	5 hepatectomies with thrombectomy; 7 hepatectomies with thrombectomy and T-tube drainage; 6 hepatectomies with resection of the common bile duct and hepaticojejunostomy; 2 liver transplantations	6 (30)	73.1	20.6	NA
Noda et al. [43]	12 (55)	NA	NA	59% were ≤5 cm; 41% were >5 cm	16 lobectomies; 6 surgically noncurative procedures	0 (0)	62	30	30
Moon et al. [44]	NA	25,280.10 ± 109,395.40	5.7 ± 5.9 mg/dL	5.8 ± 3.7	25 right hemihepatectomies ± caudate lobectomy; 4 right trisectionectomies ± caudate lobectomy; 29 left hemihepatectomies ± caudate lobectomy; 1 posterior sectionectomies; 2 anterior sectionectomies; 4 lateral sectionectomies; 2 central bisectionectomies; 1 S5-S6 bisegmentectomy; 1 isolated caudate lobectomy; 4 nonsystematic hepatectomies; 2 partial hepatectomies; 1 partial caudate lobectomy; 1 S8 subsegmentectomy	3 (4.11)	76.5	41.4	32
Oba et al. [45]	NA	2193.25 ± 2815.06	NA	6.37 ± 4.01	4 left hepatectomies; 4 right hepatectomies; 1 right hepatectomy and segment 2/3 limited resection; 1 central bisegmentectomy; 1 right trisegmentectomy; 1 anterior segmentectomy; 1 posterior segmentectomy; 6 bile duct resections and bilioenteric anastomosis	7 (53.85)	92	77	48
Wong et al. [34]	NA	50 (Ra 2–63,320)	29.05 ± 15 umol/L	9.5 ± 6	34 major hepatectomies; 3 left lateral sectionectomies	1 (2.7)	69.4	54.3	38.5
Rammohan et al. [46]	28 (71.7)	NA	6.1 ± 5.1 mg/dL	5.6 ± 3.2	16 right hepatectomies with thrombectomy; 10 extended right hepatectomies with extrahepatic bile duct excision; 9 left hepatectomies; 2 extended left hepatectomies; 2 left lateral segmentectomies	2 (5.1)	82	48	10
Ha et al. [35]	14 (100)	2043.1 ± 5528.6	5.1 ± 5.2 mg/dL	3.9 ± 1.9	13 living-donor transplantations; 1 deceased-donor transplantation	1 (7.14)	92.9	57.1	50
Kasai et al. [47]	NA	5.31 ± 13.02	1.2 ± 0.8 mg/dL	5.8 ± 3.5	41 bisectionectomies; 3 monosectionectomies; 7 combined BDRs	2 (4.54)	NA	NA	31
Chotirosniramit et al. [48]	NA	12,673.82 ± 12,499.87	11.3 ± 6.45 mg/dL	8.2 ± 4.2	2 right trisectionectomies + bile duct resection + caudate resection; 1 left trisectionectomy + bile duct resection + caudate resection; 1 right hepatectomy + bile duct resection; 4 left hepatectomies + bile duct resection; 3 right hepatectomies; 4 left hepatectomies + CBD exploration to remove BDTT; 1 left hepatectomy; 2 CBD explorations to remove BDTT and palliative biliary drainage; 1 no operation;	0 (0)	NA	60	NA
Kim et al. [36]	NA	754.25 ± 451.5	2.85 ± 1.03	NA	121 right hemihepatectomies; 7 right trisectionectomy; 81 left hemihepatectomies; 2 left trisectionectomies; 5 posterior sectionectomies; 12 anterior sectionectomies; 8 left lateral sectionectomies; 3 left medial sectionectomies; 6 central bisectionectomies; 10 nonsystematic resections; 2 liver transplantations	NA	74.5	52.9	43.6
Lin et al. [49]	NA	NA	NA	NA	25 radical resections; 7 thrombectomies through a choledochotomy; 17 palliative internal and external bile duct drainages	NA	42.86	18.37	12.24
Zhou et al. [9]	2 (28.57)	1497.03 ± 3503.20	91.36 ± 80.92 umol/L	NA	NA	NA	NA	NA	NA
Zhou et al. [17]	39 (67.24)	NA	NA	4.60 ± 1.02	36 simple hepatectomies; 11 hepatectomies plus bile duct excision	NA	NA	NA	NA
Sun et al. [50]	46 (38.33)	NA	149.4 ± 129.75	5.05 ± 2.75	19 right hepatectomies; 35 left hepatectomies; 6 left lateral sectionectomies; 13 right sectionectomies; 47 non-anatomic resections;	3 (2.5)	NA	NA	NA
Wu et al. [8]	27 (90)	NA	15.9 ± 4.15 umol/L	7.4 ± 3.05	NA	NA	NA	NA	NA
Conticchio et al.	1 (100)	2157	0.9 mg/dL	1.8	Segmentectomy S3	0 (0)	NA	NA	NA

Abbreviations: AFP, α-fetoprotein; NA, Not Available; Ra, Range; BDR, bile duct resection; CBD, common bile duct.

## Data Availability

Data supporting the results of the study will be available on request.

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
