# Peer review of "Hepatocellular Carcinoma with Bile Duct Tumor Thrombus: A Case Report and Literature Review of 890 Patients Affected by Uncommon Primary Liver Tumor Presentation"

_jcm, 2023, doi:10.3390/jcm12020423_

Round 1

Reviewer 1 Report

The authors present a case report of a patient with a HCC and a bile duct tumor thrombus. This case report was combined with a review of the literature.

1) The manuscript is structured as a case report. However, the work is labelled "review", which itself can be found in the discussion section. The manuscript needs extensive re-structuring: I think the study should primarly be organized as a review with the case report as an add-on.

2) Presentation of medical imaging needs revision. The images are too small, the MRI is not fully pictured and of poor quality.

3) The choice of treatment for the presented patient should be explained in more detail: why did the authors decide for a surgical treatment for the initial lesion, but opted for a locoregional therapy for the recurrence? Was the surgery performed minimally-invasive? If not, why?

Author Response

1) Thank you for your comment, we made a mistake submitting the paper as a review. The paper is a case report followed by an extensive narrative review in order to help colleagues in the management of a such complex clinical picture  

2) We attached more detailed images

3) The first choice of the multidisciplinary team was to try to be radical due to the young age of the patients so we performed a surgical resection in a minimally invasive approach. When the post-operative histological report showed a neoplastic biliary thrombus we aimed to treat the disease locally furthermore it was decided considering the patient's choice and peri-operative risks. 

Reviewer 2 Report

Thank you for this interesting manuscript on hepatocellular carcinoma (HCC) with bile duct tumor thrombus. 

Please expand on the hypothesized mechanisms for HCC as a thrombus in bile ducts. Compare bile duct tumor thrombus to lympho-vascular metastasis. What does "through the nerves invasion mean", and how does it lead to a tumor thrombus? Same question for "through arteriovenous shunt invasion".

Please focus the Discussion section on HCC as bile duct thrombus. COVID-19 and minimally invasive surgery should not be the lead paragraph. Please connect why HCC with bile duct tumor thrombus is of growing interest in the medical literature to the body of your literature analysis.

Author Response

Thank you for your review.

 We improved the hypothesized mechanisms for HCC as a thrombus in the bile ducts manuscript and discussion as suggested

Reviewer 3 Report

Title and content are good, and good literature and discussion. This is clinically helpful.

If there is other imaging scan performed, please include. Also, we're like to see a pathology slide for the tumor.

Author Response

thank you for your review.

We improved radiological imaging, unfortunately, pathological images are not available due to a missed consent for their publication of the patient